# Domestic Cooking Affects the Prebiotic Performances of Chinese Yam

**DOI:** 10.3390/foods11233794

**Published:** 2022-11-24

**Authors:** Lingxiao Gong, Linlin Hu, Feiyue Liu, Jingwen Chi, Rui Chen, Jing Wang

**Affiliations:** China-Canada Joint Lab of Food Nutrition and Health (Beijing), Key Laboratory of Special Food Supervision Technology for State Market Regulation, Beijing Engineering and Technology Research Center of Food Additives, School of Food and Health, Beijing Technology & Business University (BTBU), Beijing 100048, China

**Keywords:** Chinese yam, polysaccharide, in vitro fermentation, gut microbiota

## Abstract

The appropriate domestic cooking methods can retain and protect the biological properties of foods well. Thus, the objective of this study was to unravel the effect of different cooking methods on the microbiota modulatory properties of yam and their non-starch polysaccharides by an in vitro simulated digestion and fermentation model. The results showed that different cooking processes led to different changes in polysaccharide content. The polysaccharide content of yam increased by 21.3~108.2% or decreased by 12.0% compared with that of raw yam. Additionally, the soluble polysaccharides contents in all cooked yam samples significantly increased by 16.85~119.97% after in vitro digestion. The regulation of whole-yam digesta on gut microbiota was partly related with yam polysaccharides. Both yam and yam polysaccharide fermentation appeared to promote beneficial bacteria, such as *Bifidobacteria*, *Bacteroides* spp. and *Megasphaera* and suppressed bacterial pathogens such as *Ruminococcusforques* and *Escherichia-Shigella*. Household cooking significantly influenced the prebiotic performances of yam and yam polysaccharides by changing the heat-sensitive microbial substrates and their physiology properties. According to our results, normal-pressure steaming and normal-pressure boiling processes can retain the microbiota modulatory effects of Chinese yam.

## 1. Introduction

Yams (*Dioscorea oppositifolia* L.) play an important role in the diets of many countries in Asia and Africa. Chinese yam consumption has been related to a reduced risk of chronic disease such as cardiovascular diseases, diabetes, inflammation, hypertension and gastrointestinal functional disorders [1,2]. Yam polysaccharides are considered as the hotspot in yam nutrition because of their diverse bioactivities including anti-hypoglycemic, antioxidant, immune-regulatory and anti-tumor activities [3]. Recent discoveries of yam non-starch polysaccharides demonstrated their potential microbiota modulatory effects [4]. Recently, the gut microbiota has been demonstrated to be involved in the pathogenesis, progression and treatment of several chronic diseases such as cardiovascular diseases, diabetes and cancers [5,6]. The consumption of yam increased probiotic *Bifidobacteria* and *Lactobacilli* and decreased potential pathogen *Enterococcus* and *Clostridium perfringens* in antibiotic-associated diarrhea mice [7]. Additionally, the 40% ethanol extract of yam flour enriched lactose-fermenting bacteria and improved digestive capability in rats [8]. Yam with enriched resistant starch can facilitate the production of short-chain fatty acids, especially butyrate and propionate and, therefore, provide plasma cholesterol-lowering effects in rats [9]. The non-starch polysaccharides (NSPs) in yams also promoted the population of beneficial gut microbiota *Lactobacillus amylovorus*, but suppressed bacterial pathogens *Salmonella typhimurium* both in in vitro fermentation and in rat cecum [4]. Polysaccharides are considered as one of the important modifiers of the microbiota community but their microbiota modulatory effects are influenced by other dietary components [10,11]. Moreover, polysaccharides are associated with various health benefits including antioxidant activity and α-amylase and α-glucosidase inhibitory activity [12]. Therefore, the interplay between yam polysaccharides and microbiota might explain some of yams’ health benefits.

Yams can be consumed raw in salads, pickled, boiled or eaten as a jelly. Domestic cooking methods affect the natural bioactive components of food and are important to the health of humans [13]. The dynamic structure of plant cell walls consists of complex polysaccharides, phenolic compounds and proteins [14]. The cooking process can change the physicochemical properties of foods. During cooking, most of the compounds produced come from the Maillard reaction. Any alteration of the compounds may have an effect on the intestinal microorganisms to some extent [15]. It has been shown that different home cooking methods exert an influence on the bioactive components of yam (phenolic compounds, disaponins and allantoin). Different cooking techniques with different heat transfer media and heat transfer intensities will change the food composition in different ways [16]. The heat generated by cooking can also lead to changes in the structure of phenolic compounds, thus affecting their biological and antioxidant activities to some extent [15]. Cooking can also denature antimicrobial compounds naturally present in food or introduced through agriculture, thus altering their biological activity [17]. Our previous study indicated that the bioactive compound profiles (phenolic compounds, diosgenin and allantoin) and their bioaccessibilities as well as the bioactivities of yams are altered after different cooking methods. Cooking processes with lower temperatures, pressures and shorter times such as normal-pressure steaming and microwaving methods may be suitable for the retention of bioactive compounds and promotion of their bioaccessibilities [16]. In terms of polysaccharides, the content of resistant starch in raw yams was reduced from 33.9% to 6.9% after a boiling process [9]. Moreover, high temperatures during the cooking process resulted in the breakage and depolymerization of polysaccharides which may change their structure and biological activities [18]. However, information about the changes in non-starch polysaccharides and their microbiota modulatory effects of yams is limited. Studies on the effect of cooking conditions on the composition of the intestinal microbiota are scarce [15]. Hence, the present study was undertaken to investigate the changes in NSPs subjected to seven different household cooking methods. Furthermore, the gut microbiota modulatory effects of cooked yams and their NSPs were evaluated using in vitro digestion and fermentation protocols. It is worth noting that additives and seasonings are widely used during domestic cooking, which may also affect the biological activity of cooked food. In order to evaluate the changes in modulatory effects after processing with different cooking methods under different conditions, the yam samples were cooked without additives or seasonings.

## 2. Materials and Methods

### 2.1. Preparation of Cooked Yam

Fresh yam tubers (Tiegun yams harvested from Wenxian County, Henan Province, China, in October) were washed, peeled, cut and cooked according to the household cooking process of a universally acknowledged recipe. Normal-pressure steaming (NPS), high-pressure steaming (HPS), normal-pressure boiling (NPB), high-pressure boiling (HPB), stir-frying (F) and microwaving (M) were performed by our previous study [16]. Specifically, NPS was conducted under 101 kPa for 20 min and 1 L of water was used to generate the steam; HPS was conducted under 180 kPa and 117~118 °C for 15 min and 1 L of water was used to generate the steam; NPB was conducted under 101 kPa and 100 °C for 20 min, fully dipped in 1 L of cold water; HPB was conducted under 180 kPa and 117~118 °C for 15 min, fully dipped in 1 L of cold water; stir-frying was conducted under 160 °C for 15 min with 200 g/20 mL plant oil; and microwaving was conducted under 800 W for 8 min. The cooked yams were collected, freeze dried and ground into power for further study. 

### 2.2. Preparation and Purification of Yam Polysaccharides from Cooked Yams 

The preparation of crude yam NSPs was carried out according to the method described by Zhu et al. (2018) [19] with some modifications [2]. Briefly, dried yam powder was refluxed with 95% ethanol to remove lipophilic fraction. The dried residue was extracted by ultrasonic-assisted water extraction in an ultrasonic processor (KQ-700GVDV, Kunshan Ultrasonic Instrument Co., Ltd., Kunshan, China). The extractions were conducted with a solid–liquid ratio of 20:1 (*v*/*w*, mL/g), at 60 °C, with ultrasonic power at 700 w for 50 min. The extracts were purified by removing starch and proteins with α-amylase and protease under the optimal pH and temperature for 1 h, respectively. The supernatants were obtained after centrifugation at 1790× *g* for 10 min. Then, the supernatants were precipitated by adding pure alcohol (*v*:*v* = 1:4) and kept at 4 °C overnight. The sediments were collected and dissolved with distilled water and dialyzed with a 1 kDa dialysis membrane. Finally, the sediments, namely polysaccharides, were concentrated at 50 °C with rotary evaporation and lyophilized. The dried crude polysaccharides were stored until the purification and in vitro fermentation [20]. 

The polysaccharides were purified according to the method of Wang et al. (2019) [20]. Briefly, crude yam polysaccharides dissolved in distilled water were loaded onto a DEAE-52 column. The column was eluted with distilled water and 0.1, 0.3 and 0.5 mol/mL of NaCl, and the eluents were collected. The collected fractions (P20, P40 and P60) were dialyzed with a 1 kDa dialysis membrane and concentrated at 50 °C with rotary evaporation and lyophilized. The dried fractions were stored until molecular weight analysis.

### 2.3. In Vitro Fermentation of Cooked Yams and Yam Polysaccharides

Before fermentation, the cooked yams were digested under simulated saliva, gastric and intestinal digestion. The in vitro digestion and fermentation were performed according to the procedure described in our previous work with slight modification [21]. In brief, the cooked yam was successively mixed with simulated saliva, gastric electrolyte solution and intestinal enzyme fluid to mimic the digestion in oral, stomach and intestine environments. The mixtures were incubated at 37 °C for 2 min for the oral enzymatic digestion; at 37 °C for 2 h with pH 3.0 for the gastric enzymatic digestion; and at 37 °C for 2 h with pH 7.0 for the intestinal enzymatic digestion. After the intestinal digestion, the digesta was dialyzed with a 1 kDa dialysis membrane against NaCl for 48 h at 37 °C. The retentates, namely yam digesta, were freeze dried and used as substrate for fermentation. 

For in vitro fermentation, cooked yam digesta or yam polysaccharides (at the concentration of 1%, *w*/*v*) were added into the pre-sterilized basal anaerobic medium and fecal inoculum and fermented at 37 °C for 24 h in an anaerobic workstation (DG250, Don Whitley Scientific Ltd., Bingley, UK). The basal medium without cooked yam digesta or polysaccharides was prepared as a blank control. Each experiment was performed in triplicate. Stool fecal samples were collected from one male and two females between the ages of 18–25 years, with a healthy and normal diet, not taking antibiotics for the last three months, and without a history of bowel disease. All stool samples were collected in anaerobic boxes with the air removed and transported in ice boxes and then inoculated in anaerobic workstations (10% H_2_, 10% CO_2_ and 80% N_2_) at 37 °C. All operations were performed within 2 h of collection. 

### 2.4. Gut Microbiota Analysis

Total microbial DNA was extracted using the E.Z.N.A.^®^ Stool DNA Kit (Omega Bio-tek, Norcross, GA, USA) according to manufacturer’s instructions. The 16S rDNA gene sequencing was conducted on the Illumina MiSeq platform (Illumina, San Diego, CA, USA). The standard bioinformatics data were processed using the Quantitative Insights Into Microbial Ecology (QIIME, v1.8.0) pipackage. In order to obtain reliable and accurate clustering and subsequent analysis data, it is necessary to standardize the target sequence obtained after detection, and then perform OTU clustering/denoising and species classification analysis based on the standardized effective data, and form the species abundance spectrum of OTU and other species classification levels. The remaining high-quality sequences were clustered into operational taxonomic units (OTUs) with a 97% similarity using UPARSE (www.drive5.com, accessed on 27 November 2019). The taxonomic classification was carried out using a three-fifths (60%) majority rule by RDP classifier (Version 2.2, http://rdp.cme.msu.edu/, accessed on 27 November 2019). 

### 2.5. Determination of Polysaccharide Content in Yams

The water-soluble polysaccharide content in yams was determined using the phenol-sulfuric acid method of Zhu et al. (2016) [22]. One milliliter of 6% phenol solution was thoroughly mixed with 0.1 mL polysaccharide sample before being reacted with 7 mL concentrated sulfuric acid at room temperature for 20 min. The absorbance wavelength was read at 490 nm. The polysaccharide content was expressed as mg glucose/g yam. 

### 2.6. Determination of Polysaccharide Molecular Weight

The molecular weight was determined based on the method of Wu et al. (2016) [23]. Mobile phase: pure water, the 2.0 mg/mL glucan standard solution (molecular weights were 5, 25, 50, 80 and 150 KD, respectively) was prepared, and the logMw of the standard molecular weight was used as the ordinate, and the retention time of the peak was used as the abscissa to determine the molecular weight distribution of polysaccharides in the tested sample. The samples were filtered with a 0.45 μm membrane before injection. The mobile phase was pure water with a flow rate of 0.8 mL/min and the temperature was 30 °C.

### 2.7. Evaluation of Probiotic Effects of Yam Powder

In order to quantitatively compare the probiotic functions of cooked yam, the Prebiotic Index (PI) is usually calculated [21]. The PI value represents the relationship between the growth of “beneficial” bacteria such as *Bifidobacterium* and *Lactobacillus* and the growth of “less desired” bacteria such as *Clostridium* and *Escherichia coli* as a percentage of the total number of bacteria, as shown in the formula
(1)$$\mathrm{PI}=-\frac{Esc.}{Total}+\frac{Lac.}{Total}-\frac{Clos.}{Total}$$

In the formula, *Bif.*, *Esc.*, *Lac.* and *Clos.* represent the ratios of *Bifidobacterium*, *Escherichia*, *Lactobacillus* and *Clostridium* populations after fermentation for 24 h to the corresponding genus populations before fermentation. *Total* is the ratio of the relative abundance of bacteria after 24 h of fermentation to the relative f of bacteria before fermentation. 

### 2.8. Statistical Analysis

All experiments were performed in triplicate. The data were expressed as mean ± standard deviation and analyzed by one-way analysis of variance (ANOVA) using SPSS 17.0 (SPSS Inc., Chicago, IL, USA). Significant differences between sample mean values were determined by Tukey’s multiple comparison test at a 95% confidence level.

## 3. Results and Discussion

### 3.1. Effect of Household Cooking Process on Yam Polysaccharides

The polysaccharide content in uncooked raw yam was 77.4 mg/g (Table 1). This is consistent with previous result that the total water-soluble polysaccharides content in Chinese yams was in the range of 51.9~85.5 mg/g [7,24]. Cooking processes induced different changes in the polysaccharide content (Table 1). Among all the processes, NPS, HPS and microwave processes increased the polysaccharide content by 21.3~108.2% from that of the uncooked yam. However, stir-frying caused decreases in polysaccharide content by 12.0%. This is consistent with the previous research results, because the raw material surface was in contact with high-heat oil, resulting in a small amount of polysaccharide degradation [25]. The soluble polysaccharide content after NPB and HPB had no changes. A previous study on three varieties of rice found that boiling and pressure cooking did not change the total non-starch polysaccharide content in *Doongara* and *Inga*, excluding *Japonica*. However, both boiling and pressure cooking induce a re-distribution in the soluble and insoluble fractions of the total non-starch polysaccharide which, in turn, results in significant increases in soluble non-starch polysaccharide content from the uncooked samples in all the three varieties of rice [13]. The soluble polysaccharide content increases as a result of the breakage of glycosidic linkages and of weak bonds among polysaccharide chains during high-temperature heat treatment [14]. In terms of the food matrix, cooking processes are ways to change food matrix structures, which may influence the release, conversion and absorption of food components during digestion. However, the study of impacts of heat treatment on yam structure and components is limited. For quinoa, steaming had a stronger effect on losing the structural integrity compared with other heat treatments [26]. On the other hand, as shown in Table 1, the soluble polysaccharide contents in all yam samples significantly increased after in vitro digestion. The content increased by 16.85~119.97% compared to that of the undigested yam, with the greatest increase in stir-fried yam and the lowest increase in normal-pressure steamed yam. After digestion, the rank order of polysaccharide content in the cooked yam was NPS > NPB ≈ HPS > HPB ≈ Microwave > uncooked ≈ stir-frying. Generally, most of natural non-starch plant polysaccharides were not digested during human upper gastrointestinal enzymatic digestion [27]. However, the polysaccharides were very sensitive to the acidic pH in the gastric condition. Therefore, molecular weight reduction can easily occur during gastric digestion [28]. It is reported that the average molecular weights of *Artocarpus heterophyllus* Lam. polysaccharides were significantly reduced from 1485.11 kDa to 6.04 kDa after in vitro saliva, gastric and intestinal digestion [29]. The results in Table 1 show that polysaccharides with an average molecular weight more than 130 kDa were dismissed after in vitro digestion. After digestion, the distribution of the averaged molecular weight, which included 17.48~20.52% of low molecular weights (200 Da–10 kDa) and 79.48~82.52% of high molecular weights (10 kDa~130 kDa), was similar among all the cooked yam polysaccharides. Thus, the breakdown of the insoluble long-chain polysaccharides, which occurs during digestion, increases the water-soluble fraction. In this study, three purified fractions (P20, P40 and P60) were further separated by a DEAE-52 column with a 0.1 mol/L, 0.3 mol/L and 0.5 mol/L NaCl elution. The molecular weight of purified polysaccharides was 21.3 kDa for P20, 40.6 kDa for P40 and 61.1 kDa for P60. Similarly, three water-soluble polysaccharides with molecular weights of 22 KDa, 41 KDa and 23 KDa were purified from Chinese Huaishan yams [29]. 

### 3.2. Effect of Yams and Yam Polysaccharides on Gut Microbiota Communities 

Under our processing conditions, statistically significantly differences were observed in gut microbiota modulatory effects among different domestically cooked yams (Figure 1). Cooked yam digesta was effective in increasing Firmicutes and Actinobacteria at the phylum level, but inhibited or even eliminated the growth of phyla Bacteroidetes, Fusobacteria and Proteobacteria. After 24 h of fermentation in the blank group, the relative abundance of Firmicutes decreased from 63.6% to 34.5% but the relative abundance of Actinobacteria increased from 0.57% to 0.84%. Moreover, the relative abundance of Firmicutes increased from 34.5% to 97.6% (HPSY group)~99.4% (MY group), which was in accordance with a previous study [30]. However, the relative abundance of Actinobacteria either increased or decreased after fermentation with cooked yam digesta, e.g., the relative abundance of the RY group increased from 0.84% to 0.97%, but the relative abundance of the MY group decreased from 0.84% to 0.15%. A recent in vivo study revealed that treatment of 0.1% Chinese yam extract significantly decreased the Actinobacteria and Firmicutes phyla in rainbow trout [31]. The study of Meng et al. (2019) unraveled that adding Chinese yam peel to the diet significantly increased Fusobacteria and Bacteroidetes but decreased Proteobacteria phyla in fish [32]. However, few reports to date have elucidated the gut microbiota modulatory effect of whole yams.

At the genus level, yam digesta fermentation resulted in significant increases in Megamonas, Megasphaera and Dialister, but decreases in Bacteroides, Subdoligranulum, Agathobacter, Prevotella, Phascolarctobacterium, Lachnoclostridium, Ruminococcus and Bilophila. Nirmalkar et al. (2018) [33] showed that obesity is positively associated with total cholesterol and Ruminococcus. *Bilophila* thrives in the gut, which is rich in bile acids, and a high-fat diet can boost its share of the gut microbiota, increasing the risk of inflammatory bowel disease and hepatobiliary disease [34]. At the species level, the relative abundances of Megasphaera, Veillonellaceae and Megamonas were increased by over 50% after yam digesta fermentation. However, Ruminococcus, Bacteroides vulgatus and Bacteroides coprocola_DSM were significantly reduced. However, the in vivo results of Zhang et al. (2019) [7] demonstrated that the levels of *Bacteroides* spp. and *Clostridium perfringens* was increased after treatment with Chinese yam for 10–15 days in diarrhea mice, which was inconsistent with this study. The reason for the differences between the two studies with regard to microbiota resources could be that in this study, the microbiota were of healthy individuals, which was different from those of diarrhea mice. 

In terms of cooked yam polysaccharides, statistically significantly differences were observed in gut microbiota modulatory effects among different domestic cooking processes. (Figure 1). Compared with NC0, yam polysaccharide fermentation reduced the relative abundance of Firmicutes by 12.2–32.3%, and increased Actinomycetes by 18–28%. Compared with the NC24 group, the relative abundance of Firmicutes in all the yam polysaccharide groups showed an increasing trend except for the RYP group, which decreased from 34.5% to 31.3%, and the HPSYP group increased the most: to 51.4%. The relative abundance of Actinobacteria increased from 0.84% to 18.6% (NPCYP group) to 28.6% (NPSYP group), but inhibited or even eliminated the growth of phyla Fusobacteria and Proteobacteria. The proliferation of Proteobacteria can lead to an imbalance in gut microbiota and low-grade inflammation [35]. However, the Bacteroidetes phylum was not decreased after polysaccharide fermentation, which was different from digesta fermentation. The genome of Bacteroidetes has been reported to contain a large number of genes related to the acquisition and metabolism of polysaccharides [36]. Bacteroidetes, Firmicutes and Proteobacteria are three important phyla that degrade complex and indigestible polysaccharides [37]. In addition, the gut microbiota is closely associated with diseases related to glucose metabolism, such as obesity, diabetes and malnutrition, and the ratio of Firmicutes and Bacteroidetes in the gut is positively correlated with obesity [38]. Previous studies have shown that Lycium barbarum polysaccharides significantly increased the relative abundance of Firmicutes [39]. At the genus level, yam polysaccharide fermentation reduced or even eliminated *Parabacteroides*, *Lachnoclostridium*, *Blautia*, *Fusobacterium* and *Escherichia-Shigella*. These genera are closely related to the occurrence of certain diseases. For example, *Fusobacterium* is associated with the pro-inflammatory characteristics of mouse and human tumors, which can promote tumorigenesis [40]. Meanwhile, the relative abundances of *Bifidobacterium, Prevotella_9* and *Megasphaera* increased by 17.5~27.8%, 9.2~30.1% and 4.7~15.2% after polysaccharide fermentation, respectively. The relative abundance of *Bifidobacterium* increased from 0.58% to 18.1% (MYP) ~28.4% (NPSYP); the relative abundance of *Prevotella_9* increased from 0.015% to 9.2% (HPSYP group) ~30.1% (OYP group); the relative abundance of *Megasphaera* increased from 2.3% to 7.0% (OYP group) ~17.5% (HPSYP group). At the species level, polysaccharide fermentation reduced the abundance of *Ruminococcus* and *Bilophila*, but increased the relative abundances of *Bifidobacterium adolescentis* and *Bifidobacterium longum* sub sp. *longum*. The relative abundance of *Bifidobacterium adolescentis* and *Bifidobacterium longum* sub sp. *longum* increased from 0.24% to 0.98% (MYP group) ~15.2% (NPSYP group) and from 0.13% to 4.8% (HPCYP group) ~12.1% (FYP group), respectively. The effect of Chinese yam polysaccharides on the gut microbiota of rats had been studied by Kong et al. (2009) [4]. The cecal microbial communities of rats after in vitro fermentation under the presence of four different yam polysaccharides (CYP1, CYP2, CYP3, CYP4) or supplemented with CYP3 were analyzed with Denaturing Gradient Gel Electrophoresis Image analysis. All four of the yam polysaccharides increased *Lactobacillus amylovorus* and *Ruminoccus forques* but decreased *Staphylococcus aureus* and *Lactobacillussalivarius*. Interestingly, CYP1 and CYP2 increased *Salmonellatyphimurium* but decreased *Escherichia coli* O157:H7. The supplementation of CYP3 for 7 days increased bacterial species including *Lactobacillus amylovorus* but eased *Salmonella typhimurium* in rat cecum. Moreover, high doses of CYP3 increased *Lactobacillus salivarius* and *Bacillussubtilis*, but decreased *Ruminoccus forques* and *Escherichia coli* O157:H7. It was worth noting that the in vitro and in vivo effects of CYP on *Ruminoccus forques* were inconsistent. There are a few works of research whose centers of attention are on the modulatory impact of yam polysaccharides on intestine microbiota. Wu et al. (2021) [41] found that polysaccharides from loquat leaves could enhance the development of health-promoting gut microbiota such as *Megasphaera* and *Megamonas*. The results indicated that yam polysaccharide fermentation appeared to enrich beneficial bacteria as *Bifidobacteria* and *Bacteroides* spp. and suppressed bacterial pathogens such as *Ruminococcusforques* and *Escherichia-Shigella*, which was in accordance with previous study results [7,42]. The modulatory effects of yam polysaccharides on the gut microbiota depends on the structure, dosage, microbiota resources and cultured environment. 

Particularly, the household cooking processes changed the modulatory effects of yam digesta and polysaccharides. The NPSYP, HPSYP and FYP had higher relative abundances of phylum Actinobacteria compared with other cooked yam polysaccharides. In terms of Bacteroidetes, the abundance after NPSYP and HPSYP fermentation decreased by 6–10%, but increased by 7.8% after raw yam polysaccharide fermentation. The increase in Bacteroides could promote the secretion of more lyase and glycoside-bond hydrolase, promoting the degradation of intestinal polysaccharides [43]. At the genus level, NPSY, NPBY and HPSY had a relatively stronger ability to promote an increase in *Megashpaera* (average increases of 47%). Moreover, NPSYP and HPSYP promoted increases in *Bifidobacterium* by 28.32% and 23.5%. *Bifidobacterium* has been identified as one of the important probiotics in the human gut, and can protect the host from infection by enteric pathogens by producing acetate [44]. NPBY and HPBYP promoted increases in *Prevotella _9* by 26.33% and 23.98%. At the species level, NPBYP and HPBYP exhibited a stronger ability to promote the growth of *Bacterium Prevotella _9*. NPSYP, NPBYP and FYP had superiority in promoting *Bifidobacterium longum* subsp. *longum*. Previous studies demonstrated that cooking processes alter the physicochemical properties of plant foods in ways that could impact the gut microbiota. Consuming raw versus cooked tubers (potato, corn, carrot and pea) led to a lower bacterial abundance and a rise in the relative proportion of Bacteroidetes, a phylum with broad capabilities for glycan degradation. In addition, the cooking process significantly reduced heat-sensitive antimicrobial compounds such as 4-hydroxycinnamic acid, ferulic acid and vanillic acid among several additional compounds known to have significant antimicrobial effects; these and others may act synergistically to affect intestinal bacterial physiology, impairing the physiology of intestinal microbes [17]. Cooking changed physical properties of polysaccharides such as water solubility and degradation which may result in the different gut microbiota metabolization [14]. 

### 3.3. Correlation Analysis

The correlation analysis of the microbiota modulatory effect between yam polysaccharides and yam digesta is shown in Figure 2. The bacteria with abundances more than 0.11% were selected. Both yam digesta and yam polysaccharides increased the abundance of *Megamonas*, *Megasphaera*, *Veillonellaceae*, *Bifidobacterium* and *Dialister,* but decreased that of *Bacteroides*, *Escherichia-Shigella*, *Parabacteroides*, *Bilophila*, *Lachnoclostridium*, *Fusobacterium* and *Ruminococcaceae UCG-014*. The findings of this study were in accordance with the results of Wang et al. (2020) [31], in which the abundance of genus *Ruminococcaceae* was significantly decreased in 0.1% yam extract-treated rainbow trout. However, polysaccharides played a more significant role in modulatory effects on the bacteria. At the genus level, there were significant differences between the microbiota after cooking treatments of both yam and yam polysaccharides. However, the relative abundance of *Bifidobacterium* increased after yam treatment. *Bifidobacteria* are considered important components of the intestinal microbiota and possess many benefits [7]. Yam polysaccharides increased the relative abundance of *Megamonas*, *Megasphaera* and *Veillonellaceae*. 

In addition, the NPSYP, HPSYP and MY groups of yam polysaccharides had the largest increase in *Megamonas*, while the NPSY, HPSY, MY and FY groups of yam digesta had the most significant increase. In the yam polysaccharide NPSYP, HPSYP and FYP groups, *Bifidobacterium* was the dominant genus, while the contents in yam digesta HPSYP and RY groups were relatively high. For *Megasphaera*, the increase rates of yam polysaccharide NPSYP, HPSYP and HPBY groups were the highest, and the increase effects of yam digesta NPSY, NPBY and HPBY groups were obvious. Thus, the regulation of whole-yam digesta in the gut microbiota was partly related with yam polysaccharides. 

### 3.4. Function Prediction of PICRUSt

Based on the microbial community structure, the metagenomic functional characteristics after 24 h fermentation were determined using PICRUSt, as shown in Figure 3. Compared to taxonomic profiles, the functional profiles of all samples were much more similar to each other. From the level of gene abundance, it can be seen that most of the microbial activities are related to carbohydrate transport and metabolism, cell wall/membrane/capsule biogenesis, and inorganic ion transport and metabolism. Nucleotide transport and metabolism, coenzyme transport and metabolism, and lipid transport and metabolism are less involved. The abundance of polysaccharides was higher than that of yam digesta with respect to RNA processing and modification and carbohydrate transport and metabolism as well as defense mechanisms. Studies have shown that some carbohydrate polymers, usually oligosaccharides or polysaccharides, have great potential as inducers of plant defense [45]. Fang et al. (2019) [38] found that the changes in gut microbiota and the secretion of intestinal-related hormones have a great influence on glucose metabolism. The relationship between gut microbiota and polysaccharides is considered to be an important factor in the influence of polysaccharides on glucose metabolism diseases.

### 3.5. Analysis of the Probiotic Effect of Yam Powder with Different Processing Methods

Table 2 shows the PI values of yam and yam polysaccharide digesta cooked with different processing methods. The PI value is a more accurate indicator because it takes into account not only the bifidobacterial effect, but also the growth of non-probiotic strains. This allows for a relative quantification of the beneficial effect, which is more accurate than a purely qualitative evaluation and shows the growth of key bacterial groups during fermentation. The results show the modulatory effects of yams on the key “beneficial” and ”less-needed” bacteria [46]. It is obvious from the table that the prebiotic effect of yam polysaccharides is much higher than that of yams. Among all the yam polysaccharides, the PI value of the NPSYP group was the highest, followed by HPBYP and RYP. The PI values of the other cooked yam polysaccharides are lower than the polysaccharides from raw yams. Correspondingly, all the PI values of the cooked yam digesta are lower than those of the raw yam digesta, except for NPSY. 

## 4. Conclusions

The present study was carried out to compare the gut microbiota modulatory effects of raw and cooked yams as well as their polysaccharides on microbiota after in vitro gastrointestinal digestion. The results indicated that both yam and yam polysaccharide fermentation appeared to promote beneficial bacteria such as *Bifidobacteria*, *Bacteroides* spp. and *Megasphaera* and suppressed bacterial pathogens such as *Ruminococcusforques* and *Escherichia-Shigella*. Household cooking significantly influenced the prebiotic performances of yams and yam polysaccharides. Improper cooking processes might decrease the prebiotic effects of raw yams and yam polysaccharides. According to our results, normal-pressure steaming and normal-pressure boiling processes can retain the microbiota modulatory effects of Chinese yams. Our study suggested that the regulation of whole-yam digesta on gut microbiota was partly related with the yam polysaccharides. However, the prebiotic properties of yam digesta is not directly associated with the polysaccharide content. Further study should be conducted to elucidate the polysaccharide structures that influence the microbiota modulatory effects of cooked yams.

## Figures and Tables

**Figure 1 foods-11-03794-f001:**
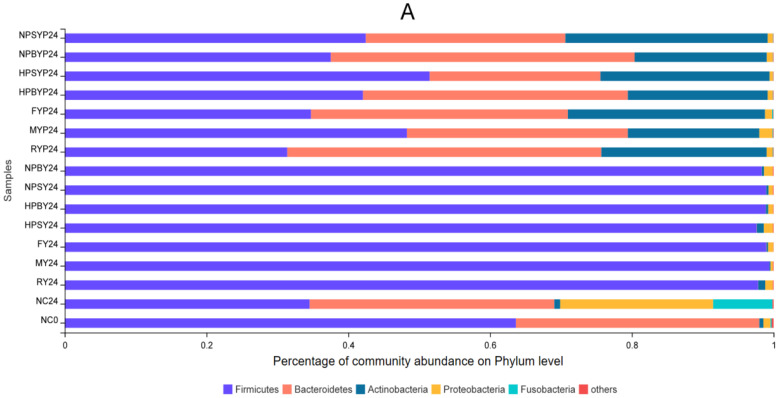
Relative bacterial abundance (at the phylum/genus/species level) after in vitro fermentation of the cooked yam digesta and polysaccharides. Note: A = at phylum level, B = at genus level, C = at species level. Note: 0 h fermentation broth, 24 h fermentation broth. NC = blank, NPSY = Normal-pressure steamed yam, NPBY = Normal-pressure boiled yam, HPSY = High-pressure steamed yam, HPBY = High-pressure boiled yam, MY = Microwaved yam, FY = Stir-fried yam. RY = Raw yam. NPSYP = Normal-pressure steamed yam polysaccharide, NPBYP = Normal-pressure boiled yam polysaccharide, HPSYP = High-pressure steamed yam polysaccharide, HPBYP = High-pressure boiled yam polysaccharide, MYP = Microwaved yam polysaccharide, FYP = Stir-fried yam polysaccharide, RYP = Raw yam polysaccharide.

**Figure 2 foods-11-03794-f002:**
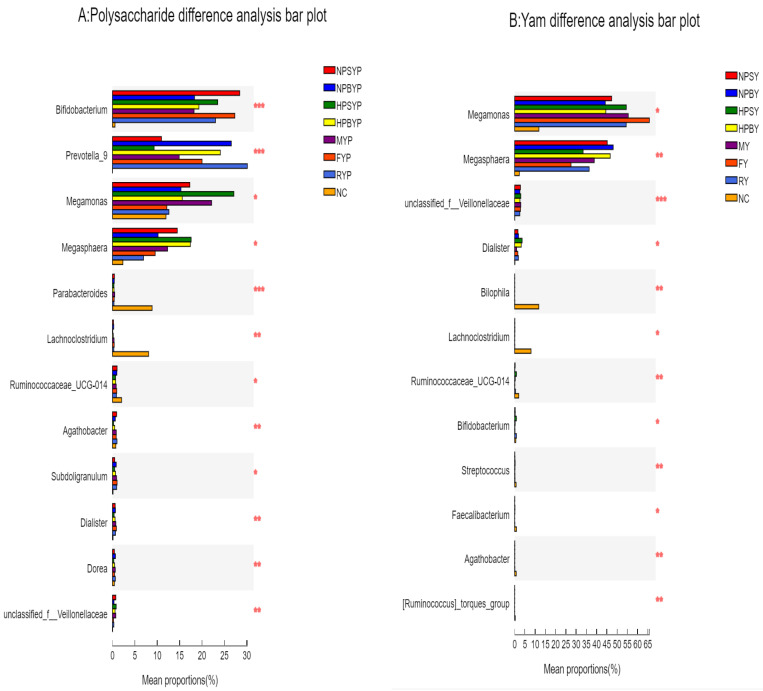
Bar chart of multi-species difference test. Note: A = Polysaccharide difference analysis bar plot, B = Yam difference analysis bar plot. The *Y* axis represents the species name at a taxonomic level, the *X* axis represents the average relative abundance in different groups of species, and the columns with different colors represent different groups. The rightmost side shows the *p* value, * 0.01 < *p* ≤ 0.05, ** 0.001 < *p* ≤ 0.01, *** *p* ≤ 0.001. NC = blank, NPSY = Normal-pressure steamed yam, NPBY = Normal-pressure boiled yam, HPSY = High-pressure steamed yam, HPBY = High-pressure boiled yam, MY = Microwaved yam, FY = Stir-fried yam. RY = Raw yam. NPSYP = Normal-pressure steamed yam polysaccharide, NPBYP = Normal-pressure boiled yam polysaccharide, HPSYP = High-pressure steamed yam polysaccharide, HPBYP = High-pressure boiled yam polysaccharide, MYP = Microwaved yam polysaccharide, FYP = Stir-fried yam polysaccharide, RYP = Raw yam polysaccharide.

**Figure 3 foods-11-03794-f003:**
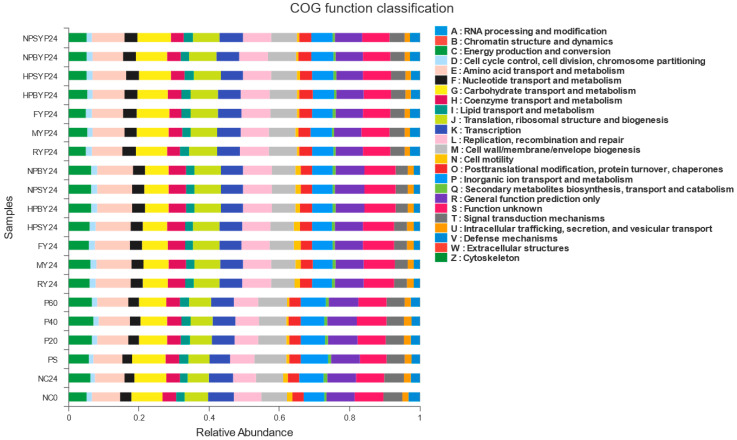
Histogram of COG function classification statistics. Note: 0 h fermentation broth, 24 h fermentation broth. NC = blank, NPSY = Normal-pressure steamed yam, NPBY = Normal-pressure boiled yam, HPSY = High-pressure steamed yam, HPBY = High-pressure boiled yam, MY = Microwaved yam, FY = Stir-fried yam. RY = Raw yam. NPSYP = Normal-pressure steamed yam polysaccharide, NPBYP = Normal-pressure boiled yam polysaccharide, HPSYP = High-pressure steamed yam polysaccharide, HPBYP = High-pressure boiled yam polysaccharide, MYP = Microwaved yam polysaccharide, FYP = Stir-fried yam polysaccharide RYP = Raw yam polysaccharide. PS: glucan standard (The molecular weight: 80 KD); P20: The molecular weight: 20 KD; P40: The molecular weight: 40 KD; P60: The molecular weight: 60 KD.

**Table 1 foods-11-03794-t001:** The content and molecular weight of water-soluble yam polysaccharides.

			NPSY	NPBY	HPSY	HPBY	MY	FY	Raw
Content (mg/g) *	Undigested		161.4 ± 0.6 ^e^	80.2 ± 0.9 ^b^	100.9 ± 6.5 ^d^	83.9 ± 1.8 ^b^	93.9 ± 2.1 ^c^	68.1 ± 5.5 ^a^	77.4 ± 4.7 ^b^
	Digested		188.6 ± 2.1 ^e^	175.8 ± 7.7 ^d^	172.1 ± 1.9 ^cd^	162.1 ± 7.9 ^bc^	162.2 ± 4.8 ^bc^	149.8 ± 0.9 ^a^	154.9 ± 8.7 ^ab^
Molecular weight **	Undigested	200 Da–130 kDa	53.23%	65.82%	49.11%	62.37%	67.88%	26.98%	44.66%
		>130 kDa	46.66%	34.18%	50.89%	37.63%	32.12%	73.02%	55.34%
	Digested	200 Da–10 kDa	18.03%	17.48%	18.13%	17.59%	17.86%	17.85%	20.52%
		10 kDa~130 kDa	81.97%	82.52%	81.87%	82.41%	82.14%	82.15%	79.48%

Note: NPSY = Normal-pressure steamed yam, NPBY = Normal-pressure boiled yam, HPSY = High-pressure steamed yam, HPBY = High-pressure boiled yam, MY = Microwaved yam, FY = Stir-fried yam. The same letter indicates not significant difference (*p* > 0.05). * The content was expressed as mg glucose/g yam sample. ** The molecular weight was expressed as the percentage of polysaccharide content within the range.

**Table 2 foods-11-03794-t002:** The PI values of yams and yam polysaccharides.

PI			
NPSYP	121.13	NPSY	30.47
NPBYP	55.67	NPBY	8.37
HPSYP	86.08	HPSY	−8.07
HPBYP	107.84	HPBY	−2.78
FYP	60.40	FY	−11.84
MYP	75.40	MY	5.11
RYP	100.81	RY	16.77
Control			
NC	−118.07		

## Data Availability

Data is contained within the article.

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
