# Peer review of "Domestic Cooking Affects the Prebiotic Performances of Chinese Yam"

_foods, 2022, doi:10.3390/foods11233794_

Round 1
Reviewer 1 Report
The manuscript addresses a very interesting and understudied topic, i. e., the effects of cooking methods on the gut microbiota. The overall approach is sound, i. e. the in vitro digestion and fermentation of yams and yam polysaccharides followed by determination of gut microbiota composition. However, the methods employed for microbiota analysis are not correctly explained and seem inappropriate. The quality of the writing needs serious improvement, especially in the Results and Discussion. Overall, the manuscript needs substantial work, in order to really leverage the large amount of work performed.
Some general nomenclature and format issues:
-please do not use “flora” when referring to the microbiota.
-Genus and species names need to be written in italics.
-The terms prebiotic and probiotic are often confused in the text.
-There is no line numbering of the text, which makes review more difficult.
Methods:
-it is important to provide more details regarding the fecal inoculum used. Was it prepared from a pool of fecal samples from different individuals? What was the age range, health status, diet, etc of the individuals? Were the fecal samples or the inoculum frozen before use in the fermentations?
-Sequence processing: basic information needs to be provided regarding sequence processing and quality control before analysis. What programs and parameters were used?
-No information is provided regarding basic issues in 16S rRNA data analysis, such as normalization of data prior to further analysis.
-The Prebiotic Index presented is problematic and is not in general use. First of all, Clostridium is a very broad genus containing a multitude of very different species that have widely different roles and potential effects on the host. It is therefore not at all advisable to use it as proposed. Moreover, Lactobacillus and Clostridium are not even detected according to the graphs presented. Importantly, the authors state that the index takes into account “the ratio of the total bacterial population after fermentation for 24 h to the total bacterial population before fermentation.” How can they know the total bacterial population? !6S data are relative and provide no information whatsoever regarding total abundance of bacteria.
-The statistical analysis is not adequate for microbiome data. Differential abundance analyses specifically designed for the characteristics of microbiome data are required. These need to take into account the specificities of microbiome data such as compositionality, i. e. the fact that data are relative and the proportions of all taxa must add to one, so that they are not free to vary independently from one another. Specific programs that deal with this issue have been developed for microbiome analyses, such as ANCOM-BC.
-The way in which PiCrust was applied is not described in the Methods.
Results and discussion:
-In the species level data presented, most of the groups fail to be assigned to a species, and are therefore left at the “unclassified” status. Only some species are actually identified, which questions the usefulness of trying to take the analysis to this level. It is incorrect to refer to most of the groupings presented in Figure 1C as species, as in the sentences “At the species level, the relative abundances of Megasphaera, Veillonellaceae and Megamonas…” or “At the species level, polysaccharide fermentation reduced the abundance of Ruminococcus, and Bilophila…”. Megasphaera and Megamonas, as well as Ruminococcus and Bilophila, are genera and Veillonellaceae is actually a family. The groupings represented in the figure are likely unclassified OTUs belonging to these genera or family, and need to be referred to as such.
-Ruminocccaceae is not a genus but a family.
-The sentence “Bacteroidetes, Firmicutes, and Proteobacteria were three important Bacteroides that degrade complex and indigestible polysaccharides [31]” needs to be removed or corrected as obviously it is completely incorrect (for starters, Bacteroidetes, Firmicutes, and Proteobacteria are phyla and cannot belong to a genus such as Bacteroides).
-When differences in the relative abundance of bacterial groups are presented, It is necessary to always clarify the baseline they are being compared to, since sometimes they are compared to the control before fermentation (0h) and others they are compared to the control after fermentation (24h).
-Again, in the sentence “At the species level, NPBYP and HPBYP exhibited the stronger ability to promote the growth of “Bacterium Prevotella”, “Bacterium Prevotella” is not a species (or any other kind of valid taxon). The corresponding grouping represented in the figure is an OTU that matched an uncultured bacterium in the RDP database belonging to Prevotella.
-There is a section entitled “Correlation analyses” but no correlation analyses between cooked yams and polysaccharides are really reported. The statistical methodology employed in Figure 2 is not specified. All comparisons of bacterial abundance between groups need to take into account the statistical caveats already mentioned and use specialized programs such as ANCOM-BC (or others that take into account the specifics of microbiome data).
-Why is “yam surimi” suddenly mentioned in page 14?
-“The last microbial communities used to be influenced by means of the blended results of extraordinary substrates remained in yam digesta.”. This sentence is meaningless.
-The results obtained from PiCrust are not analyzed to any extent.
-The issues with the PI have already been mentioned above.
Author Response
The manuscript submitted for evaluation is interesting and is a continuation of research using “Chinese yam”, the first part of which has already been published by the authors in Food Chemistry. A few points need clarification:
- The authors expressed the polysaccharide content as mg glucose/g yam. Probably the authors have prepared a standard curve using glucose. What concentration range was used to prepare the standard solution?
Thank you very much for your comments. We first prepared a glucose solution with a concentration unit of mg/mL, and then determined the soluble polysaccharide content in yam by taking a certain amount of yam and adding a quantitative solvent, at which point the unit was converted to mg/g. The concentration range was 0.02 mg/mL to 0.1 mg/mL.
- Should the caption "percent of community abundance ..." shown under Figure 1. should not include a scale from 0 to 100%?
Thank you very much for your comments. The X-axis title is revised as “percentage of community abundance”.
- Please provide the names of the relevant antimicrobial compounds that the authors had in mind for the sentence posted - "Additionally, the cooking processes significantly decreased heat-sensitive antimicrobial compounds that impair gut microbial physiology".
Thank you very much for your comments. Multiple additional compounds with known antibacterial effects were significantly decreased in cooked tubers, including 4-hydroxycinnamic acid, ferulate and vanillic acid; these and others may act in concert to impact gut bacterial physiology. We have provided the names and explanations of the relevant antimicrobial compounds in the article, as detailed in Line 358-Line 362 of the reworked manuscript.
- The results in Table 2 should be further discussed by the authors.
Thank you very much for your comments. We have discussed the results of Table 2 in detail. For details, see Line 409-Line 419 of the reworked manuscript. The PI value is a more accurate indicator because it takes into account not only the bifidobacterial effect, but also the growth of non-probiotic strains. This allows for a relative quantification of the beneficial effect, which is more accurate than a purely qualitative evaluation and the growth of key bacterial groups during fermentation. The results show the modulatory effects of yam on the on the key ‘beneficial’ and ‘less-needed’ bacteria.
Reference: (Paesani, C.; Salvucci, E.; Moiraghi, M.; Canigia, L.F.; Perez, G.T. Arabinoxylan from Argentinian whole wheat flour promote the growth of Lactobacillus reuteri and Bifidobacterium breve. Lett. Appl. Microbiol. 2019, 68, 142-148, doi:10.1111/lam.13097.)
- In the "Conclusions" section, the authors used the phrase "significantly influenced..." i.e. what was the impact?
Thank you very much for your comments. The cooking processes might decrease the prebiotic effects of raw yam and yam polysaccharides. According to our results, normal pressure steaming process and normal pressure boiling process can retain the microbiota modulatory effects of Chinese yam. Our study suggest that the regulation of whole yam digesta on gut microbiota was partly related with the yam polysaccharide. For details, please refer to Line 428-436 of the reworked manuscript.
Reviewer 2 Report
The objective of this study was to unravel the effect of different cooking methods on the microbiota modulatory properties of yam and their non-starch polysaccharides by an in vitro simulated digestion and fermentation model.
After careful review, I can say that the presentation of results and graphs/tables are not reader friendly. In the discussion section, the authors should discuss their results with more up-to-date sources and in a more critical and rigorous manner. The writing name of the microorganisms should be checked and the names should be italicized. In the conclusion, pressure/boiling cooking can increase prebiotic activity, but what about the nutritional value? We should also consider this.
I am in the opinion that the current manuscript does not really add significantly to the literature and it is of lower priority than other manuscripts in Foods.
Since the study was conducted on a traditional food, it may be more appropriate to submit it to a local journal.
Author Response
The manuscript addresses a very interesting and understudied topic, i. e., the effects of cooking methods on the gut microbiota. The overall approach is sound, i. e. the in vitro digestion and fermentation of yams and yam polysaccharides followed by determination of gut microbiota composition. However, the methods employed for microbiota analysis are not correctly explained and seem inappropriate. The quality of the writing needs serious improvement, especially in the Results and Discussion. Overall, the manuscript needs substantial work, in order to really leverage the large amount of work performed.
- Some general nomenclature and format issues:
-please do not use “flora” when referring to the microbiota.
Thank you very much for your comments. We have replaced the word "flora" with "microbiota".
-Genus and species names need to be written in italics.
Thank you very much for your comments. We have changed the genus and species names to italics.
-The terms prebiotic and probiotic are often confused in the text.
Thank you very much for your comments. We have corrected prebiotic and probiotic.
-There is no line numbering of the text, which makes review more difficult.
Thank you very much for your comments. We have added line numbers to the text.
- Methods:
-It is important to provide more details regarding the fecal inoculum used. Was it prepared from a pool of fecal samples from different individuals? What was the age range, health status, diet, etc of the individuals? Were the fecal samples or the inoculum frozen before use in the fermentations?
Thank you very much for your comments. In this study, one male and two females between the ages of 18-25 years, with a healthy and normal diet, not taking antibiotics for the last three months and without a history of bowel disease were selected as stool samples. All stool samples were collected in anaerobic boxes with air removed and transported in ice boxes and then inoculated in anaerobic workstations (10% H2, 10% CO2 and 80% N2) at 37°C. All operations were performed within 2 h of collection. The information has been added in the revised manuscript.
-Sequence processing: basic information needs to be provided regarding sequence processing and quality control before analysis. What programs and parameters were used?
Your comments are greatly appreciated. In this study, a PCR instrument (ABI GeneAmp® Model 9700) was used for sequence processing, where PCR reaction parameters included: a. 1× (3 minutes at 95°C); b. Number of cycles× (30 seconds at 95°C; 30 seconds at annealing temperature°C; 45 seconds at 72°C); c. 10 minutes at 72°C, 10°C until halted by user. In addition, DNA purity and concentration were checked by NanoDrop2000, and DNA integrity was checked by agarose gel electrophoresis.
-No information is provided regarding basic issues in 16S rRNA data analysis, such as normalization of data prior to further analysis.
Thank you very much for your comment. The samples were analyzed by 16s rRNA, and the average sequence length was about 421.1 bp, which met the quality control requirements. The sample dilution curve of the Sobs index shows that when the curve reaches plateau, the sequencing depth is considered to cover all species. As shown by the Sobs index, the data is large enough to reflect the biological information for each sample.
-The Prebiotic Index presented is problematic and is not in general use. First of all, Clostridium is a very broad genus containing a multitude of very different species that have widely different roles and potential effects on the host. It is therefore not at all advisable to use it as proposed. Moreover, Lactobacillus and Clostridium are not even detected according to the graphs presented. Importantly, the authors state that the index takes into account “the ratio of the total bacterial population after fermentation for 24 h to the total bacterial population before fermentation.” How can they know the total bacterial population? 16S data are relative and provide no information whatsoever regarding total abundance of bacteria.
The PI value is a more accurate indicator because it takes into account not only the bifidobacterial effect, but also the growth of non-probiotic strains. This allows for a relative quantification of the beneficial effect, which is more accurate than a purely qualitative evaluation and the growth of key bacterial groups during fermentation. The results show the modulatory effects of yam on the on the key ‘beneficial’ and ‘less-needed’ bacteria. (Paesani, C.; Salvucci, E.; Moiraghi, M.; Canigia, L.F.; Perez, G.T. Arabinoxylan from Argentinian whole wheat flour promote the growth of Lactobacillus reuteri and Bifidobacterium breve. Lett. Appl. Microbiol. 2019, 68, 142-148, doi:10.1111/lam.13097.) By reading the literature, we found that there are many researchers who also use PI to evaluate the microbiota modulatory effects. (Slizewska, K. The citric acid-modified, enzyme-resistant dextrin from potato starch as a potential prebiotic. Acta Biochim. Pol. 2013, 60, 671-675; Neyrinck, A.M.; Possemiers, S.; Druart, C.; van de Wiele, T.; De Backer, F.; Cani, P.D.; Larondelle, Y.; Delzenne, N.M. Prebiotic Effects of Wheat Arabinoxylan Related to the Increase in Bifidobacteria, Roseburia and Bacteroides/Prevotella in Diet-Induced Obese Mice. PLoS One 2011, 6, 12, doi:10.1371/journal.pone.0020944; Li, T.; Lu, X.; Yang, X. Stachyose-enriched alpha-galacto-oligosaccharides regulate gut microbiota and relieve constipation in mice. J Agric Food Chem 2013, 61, 11825-11831, doi:10.1021/jf404160e.; Tiwari, D.P.; Shah, P.; Van den Abbeele, P.; Marzorati, M.; Calatayud, M.; Ghyselinck, J.; Dubey, A.K.; Narayanan, S.; Jain, M. Microbial fermentation of Fossence (TM), a short-chain fructo-oligosaccharide, under simulated human proximal colonic condition and assessment of its prebiotic effects-a pilot study. FEMS Microbiol. Lett. 2021, 368, 13, doi:10.1093/femsle/fnab147.) Therefore, we choose the PI values to evaluate the microbiota modulatory effects of different cooked yam digesta and yam polysaccharides.
For the Clostridium, some pathogenic Clostridium perfringens can reside in the host as part of the normal microbiota or as potential spores with no apparent adverse effects. However, Clostridium exotoxins can cause minor and even fatal damage, affecting the gastrointestinal tract (enterotoxins), soft tissues and organs (tissue-damaging toxins), or causing neuronal dysfunction (neurotoxins). Some Clostridium is also the pathogens of enterotoxemia, gas gangrene, necrotizing enteritis, black leg disease and black leg disease in animals. (Zaragoza, N.E.; Orellana, C.A.; Moonen, G.A.; Moutafis, G.; Marcellin, E. Vaccine Production to Protect Animals Against Pathogenic Clostridia. Toxins 2019, 11, 29, doi:10.3390/toxins11090525.; Khiav, L.A.; Zahmatkesh, A. Vaccination against pathogenic clostridia in animals: a review. Trop. Anim. Health Prod. 2021, 53, 12, doi:10.1007/s11250-021-02728-w.)
For ‘Lactobacillus and Clostridium are not even detected according to the graphs presented’. In our results analysis we only filtered out the top 20 species in terms of richness. The relative abundance of Lactobacillus and Clostridium are as the following figure. In the manuscript, the data are showed and discussed in the text rather than in the figure. If necessary, we can add the figure in the revised manuscript.
For the confused expression, we have removed this sentence, as detailed in Line187-Line 189 of the reworked manuscript.
-The statistical analysis is not adequate for microbiome data. Differential abundance analyses specifically designed for the characteristics of microbiome data are required. These need to take into account the specificities of microbiome data such as compositionality, i. e. the fact that data are relative and the proportions of all taxa must add to one, so that they are not free to vary independently from one another. Specific programs that deal with this issue have been developed for microbiome analyses, such as ANCOM-BC.
Thank you very much for your comments. Based on the need for the purpose of this study, the assay currently used is able to support the conclusions and the method is one of the common platforms that have been used for similar analyses in recent years, in addition, the cost factor was also taken into consideration, so we chose this method in this study. However, we strongly acknowledge your comments and will therefore further consider using more advanced methods to improve the analytical system in our future studies.
-The way in which PiCrust was applied is not described in the Methods.
Thank you very much for your comments. We first normalized the OTU abundance table using PICRUSt (PICRUSt process stores the COG information and KO information corresponding to the greengene id) to remove the effect of the number of copies of the 16S marker gene in the species genome. Then the COG family information and KEGG Ortholog (KO) information corresponding to OTUs were obtained from the greengene id corresponding to each OUT, and the abundance of each COG and KO abundance were calculated. Finally, according to the information of COG database, the descriptive information of each COG and its functional information were parsed from the eggNOG database to obtain the functional abundance profile, and then according to the KEGG database (Kyoto Encyclopedia of Genes and Genomes, Kyoto Encyclopedia of Genes and Genomes, http://www.genome .jp/kegg/) to obtain KO, Pathway and EC information, and the abundance of each functional class was calculated from OTU abundance. In addition, for Pathway, PICRUSt can be applied to obtain the three levels of metabolic pathway information to obtain the abundance table for each level separately. Functional prediction of KEGG, COG and Pfam for 16S sequences is conducted following the instruction in http://picrust.github.io/picrust/.
- Results and discussion:
-In the species level data presented, most of the groups fail to be assigned to a species, and are therefore left at the “unclassified” status. Only some species are actually identified, which questions the usefulness of trying to take the analysis to this level. It is incorrect to refer to most of the groupings presented in Figure 1C as species, as in the sentences “At the species level, the relative abundances of Megasphaera, Veillonellaceae and Megamonas…” or “At the species level, polysaccharide fermentation reduced the abundance of Ruminococcus, and Bilophila…”. Megasphaera and Megamonas, as well as Ruminococcus and Bilophila, are genera and Veillonellaceae is actually a family. The groupings represented in the figure are likely unclassified OTUs belonging to these genera or family, and need to be referred to as such.
Thank you very much for your comments. In our analysis of the genus-level results, we filtered out the top 20 species in terms of richness percentage, which have unclassified species, just for relative expression of such OTUs.
-Ruminocccaceae is not a genus but a family.
Thank you very much for your comments. We have corrected the issue as detailed in Line 372-Line 374 of the reworked manuscript.
-The sentence “Bacteroidetes, Firmicutes, and Proteobacteria were three important Bacteroides that degrade complex and indigestible polysaccharides [31]” needs to be removed or corrected as obviously it is completely incorrect (for starters, Bacteroidetes, Firmicutes, and Proteobacteria are phyla and cannot belong to a genus such as Bacteroides).
Thank you very much for your comments. We have removed this sentence, as detailed in Line 293 of the reworked manuscript.
-When differences in the relative abundance of bacterial groups are presented, It is necessary to always clarify the baseline they are being compared to, since sometimes they are compared to the control before fermentation (0h) and others they are compared to the control after fermentation (24h).
Thank you very much for your comments. In this study, the microbiota community is compared after 24h-fermentation with different yam digesta or polysaccharide samples. The changes of the microbiota were firstly demonstrated by comparison the community in the control group before and after 24h-fermention. And then, the modulatory effects of different yam digesta or polysaccharide samples were evaluated by comparing the sample fermentation (24h) with the control group after fermentation (24h).
-Again, in the sentence “At the species level, NPBYP and HPBYP exhibited the stronger ability to promote the growth of “Bacterium Prevotella”, “Bacterium Prevotella” is not a species (or any other kind of valid taxon). The corresponding grouping represented in the figure is an OTU that matched an uncultured bacterium in the RDP database belonging to Prevotella.
Thank you very much for your comments. The sentence should be expressed as ‘At the species level, NPBYP and HPBYP exhibited the stronger ability to promote the growth of Prevotella _9’.
-There is a section entitled “Correlation analyses” but no correlation analyses between cooked yams and polysaccharides are really reported. The statistical methodology employed in Figure 2 is not specified. All comparisons of bacterial abundance between groups need to take into account the statistical caveats already mentioned and use specialized programs such as ANCOM-BC (or others that take into account the specifics of microbiome data).
The statistical method we use in Figure 2 is the Kruskal-Wallis H test, referred to as the Kruskal rank sum test.
-Why is “yam surimi” suddenly mentioned in page 14?
Thank you very much for your comments. We have made changes here, as detailed in Line 385-Line 387 of the reworked manuscript. The reason for using “yam digesta” is that it is digested by gastrointestinal tract before fermentation. In brief, the cooked yam was successively mixed with simulated saliva, gastric electrolyte solution and intestinal enzyme fluid to mimic the digestion in oral, stomach and intestine. The mixtures were incubated at 37 ℃ for 2min for the oral enzymatic digestion, at 37 ℃ for 2 h with pH 3.0 for gastric enzymatic digestion and at 37 ℃ for 2 h with pH 7.0 for intestinal enzymatic digestion. After the intestinal digestion, the digesta was dialyzed with 1 kDa dialysis membrane against NaCl for 48 h at 37 ℃. The retentates, namely yam digesta were freeze dried and used as substrate for fermentation.
-“The last microbial communities used to be influenced by means of the blended results of extraordinary substrates remained in yam digesta”. This sentence is meaningless.
Thank you very much for your comment, we have deleted this sentence.
-The results obtained from PiCrust are not analyzed to any extent.
Thank you very much for your comments. We have corrected the issue as detailed in Line 392-Line 402 of the reworked manuscript.
-The issues with the PI have already been mentioned above.
Thank you very much for your comments. We have explained the issues involved.

Reviewer 3 Report
The manuscript submitted for evaluation is interesting and is a continuation of research using „Chinese yam”, the first part of which has already been published by the authors in Food Chemistry. A few points need clarification:
(1) The authors expressed the polysaccharide content as mg glucose/g yam. Probably the authors have prepared a standard curve using glucose. What concentration range was used to prepare the standard solution?
(2) Should the caption "percent of community abundance ..." shown under Figure 1. should not include a scale from 0 to 100%?
(3) Please provide the names of the relevant antimicrobial compounds that the authors had in mind for the sentence posted - "Additionally, the cooking processes significantly decreased heat-sensitive antimicrobial compounds that impair gut microbial physiology".
(4) The results in Table 2 should be further discussed by the authors.
(5) In the "Conclusions" section, the authors used the phrase "significantly influenced..." i.e. what was the impact?
Author Response
The objective of this study was to unravel the effect of different cooking methods on the microbiota modulatory properties of yam and their non-starch polysaccharides by an in vitro simulated digestion and fermentation model. After careful review, I can say that the presentation of results and graphs/tables are not reader friendly. In the discussion section, the authors should discuss their results with more up-to-date sources and in a more critical and rigorous manner. The writing name of the microorganisms should be checked and the names should be italicized. In the conclusion, pressure /boiling cooking can increase prebiotic activity, but what about the nutritional value? We should also consider this. I am in the opinion that the current manuscript does not really add significantly to the literature and it is of lower priority than other manuscripts in Foods. Since the study was conducted on a traditional food, it may be more appropriate to submit it to a local journal.
Thank you very much for your input. In addition, we have made profound changes in accordance with the comments of other reviewers to further improve the content of this study. Finally, articles related to this study have been recently published by leading journals related to the food field, such as.
Gut microbial fermentation promotes the intestinal anti-inflammatory activity of Chinese yam polysaccharides, Food Chemistry. 2023, 402, 11, doi:10.1016/j.foodchem.2022.134003.
Effect of Food Thermal Processing on the Composition of the Gut Microbiota, Journal of Agricultural And Food Chemistry.2018, 66, 11500-11509, doi:10.1021/acs.jafc.8b04077
Reviewer 4 Report
The manuscript of Gon et al. presents an interesting work about the potential influence of different household cooking processes in the prebiotic effects of yam.
In general, the manuscript fulfills the requirements established by Foods journal. The Introduction section clearly indicates the objectives of the work and its significance. The Materials and methods part is well-described and explained, with enough details to be reproducible. The study presents interesting results which are well discussed and could be a good starting point for further research works. The figures and tables are necessary for the presentation of the results. The references are properly presented as they are identified with numbers in square brackets in the text.
Overall, the manuscript is well-written after making some changes. Suggestions and comments to improve the quality of the manuscript are the following:
1. A concise cover letter should have been submitted with view to explaining the interest and originality of the study and, above all, the reasons for why the work should be published.
2. Please, indicate in the abstract the cooking procedures selected and studied as well as which one was responsible for the marked increase (108.2%) and reduction (12%) of the polysaccharide content in comparison with the raw yam.
3. Introduction section must be extended as it does not provide sufficient background. Please, include more relevant references.
4. Line 76: please, indicate in the objectives the 7 cooking procedures selected and studied by authors.
5. Authors must include a paragraph explaining the most important limitations of the study.
6. Conclusions section must be enriched to summarize the most important findings of the work.
Author Response
The manuscript of Gon et al. presents an interesting work about the potential influence of different household cooking processes in the prebiotic effects of yam. In general, the manuscript fulfills the requirements established by Foods journal. The Introduction section clearly indicates the objectives of the work and its significance. The Materials and methods part is well-described and explained, with enough details to be reproducible. The study presents interesting results which are well discussed and could be a good starting point for further research works. The figures and tables are necessary for the presentation of the results. The references are properly presented as they are identified with numbers in square brackets in the text. Overall, the manuscript is well-written after making some changes. Suggestions and comments to improve the quality of the manuscript are the following:
- A concise cover letter should have been submitted with view to explaining the interest and originality of the study and, above all, the reasons for why the work should be published.
Our study demonstrated that improper cooking processes might decrease the prebiotic effects of raw yam and yam polysaccharides. Among the selected processes, normal pressure steaming and normal pressure boiling were an excellent choice for cooking Chinese yam based on the standard of health. Our study also suggested that the regulation of whole yam digesta on gut microbiota was partly related with the yam polysaccharide. However, cooking process resulted in the content and structure changes of the yam polysaccharide. Our findings are the first to highlight that cooked and digesta composition instead of its original composition and the combined effects of its components should be taken into consideration when selecting yam as prebiotics.
- Please, indicate in the abstract the cooking procedures selected and studied as well as which one was responsible for the marked increase (108.2%) and reduction (12%) of the polysaccharide content in comparison with the raw yam.
Thank you very much for your comments. We have made the relevant additions in the text. Among them, the cooking conditions and procedures are added in the revised manuscrip. In addition, we believe that it may be due to the loss of reducing sugars that occurs during frying, resulting in the lowest polysaccharide content in this way and the highest polysaccharide content when steaming at room temperature. During the cooking process, high temperature affects the polysaccharide structure in many ways, mostly by breaking links and promoting depolymerization, thus increasing the polysaccharide content. (Weiwei, L.; Jingya, W.; Zhongqin, C.; Xudong, G.; Yue, C.; Zihan, X.; Qingwen, G.; Qiqi, M.; Haixia, C. Physicochemical properties of polysaccharides from Lentinus edodes under high pressure cooking treatment and its enhanced anticancer effects. International Journal of Biological Macromolecules 2018, 115, 994-1001, doi:10.1016/j.ijbiomac.2018.04.094.) There is also evidence that plant texture and cell wall composition change during cooking which may assist the digestion of foods. (Ratnayake, R.M.S.; Melton, L.D.; Hurst, P.L. Influence of cultivar, cooking, and storage on cell-wall polysaccharide composition of winter squash (Cucurbita maxima). J. Agric. Food Chem. 2003, 51, 1904-1913, doi:10.1021/jf020772+.) Further study will be conducted to elucidate the yam polysaccharide structure changes during the cooking process.
- Introduction section must be extended as it does not provide sufficient background. Please, include more relevant references.
Your comments are greatly appreciated. We have refined the preface to provide a fuller context for the study and have added additional relevant references.
- Line 76: please, indicate in the objectives the 7 cooking procedures selected and studied by authors.
Thank you very much for your comments. The cooking conditions are added in the revised manuscript. For details, please refer to Line 94-Line 100 of the reworked manuscript.
- Authors must include a paragraph explaining the most important limitations of the study.
Thank you very much for your comments. The additives and seasonings are widely used during domestic cooking, which may also affect the biological activity of cooked foods. The limitations are discussed in the revised manuscript. For details, please refer to Line 82-Line 86 of the reworked manuscript.
- Conclusions section must be enriched to summarize the most important findings of the work.
Thank you very much for your comments. We have revised this part in the revised manuscript.
Round 2
Reviewer 2 Report
After revision, the manuscript seems to better for publication. Best regards,
Reviewer 4 Report
Dear Authors,
After reviewing the last version of the manuscript, I consider that this work is worthy of publication.